# Bayesian Spatial Modelling of HIV Prevalence in Jimma Zone, Ethiopia

**DOI:** 10.3390/diseases11010046

**Published:** 2023-03-08

**Authors:** Legesse Kassa Debusho, Nemso Geda Bedaso

**Affiliations:** 1Department of Statistics, College of Science, Engineering and Technology, University of South Africa, Private Bag X6, Florida 1710, South Africa; 2Department of Statistics, College of Natural and Computational Science, Madda Walabu University, Bale Robe P.O. Box 247, Ethiopia

**Keywords:** Bayesian modelling, global Moran’s I, Getis–Ord *G*_*i*_* statistic, HIV prevalence, odds ratios, spatial clustering

## Abstract

**Background**: Although the human immunodeficiency virus (HIV) is spatially heterogeneous in Ethiopia, current regional estimates of HIV prevalence hide the epidemic’s heterogeneity. A thorough examination of the prevalence of HIV infection using district-level data could assist to develop HIV prevention strategies. The aims of this study were to examine the spatial clustering of HIV prevalence in Jimma Zone at district level and assess the effects of patient characteristics on the prevalence of HIV infection. **Methods**: The 8440 files of patients who underwent HIV testing in the 22 Districts of Jimma Zone between September 2018 and August 2019 were the source of data for this study. The global Moran’s index, Getis–Ord $G_{i}^{*}$ local statistic, and Bayesian hierarchical spatial modelling approach were applied to address the research objectives. **Results**: Positive spatial autocorrelation was observed in the districts and the local indicators of spatial analysis using the Getis–Ord statistic also identified three districts, namely Agaro, Gomma and Nono Benja, as hotspots, and two districts, namely Mancho and Omo Beyam, as coldspots with 95% and 90% confidence levels, respectively, for HIV prevalence. The results also showed eight patient-related characteristics that were considered in the study were associated with HIV prevalence in the study area. Furthermore, after accounting for these characteristics in the fitted model, there was no spatial clustering of HIV prevalence suggesting the patient characteristics had explained most of the heterogeneity in HIV prevalence in Jimma Zone for the study data. **Conclusions**: The identification of hotspot districts and the spatial dynamic of HIV infection in Jimma Zone at district level may allow health policymakers in the zone or Oromiya region or at national level to develop geographically specific strategies to prevent HIV transmission. Because clinic register data were used in the study, it is important to use caution when interpreting the results. The results are restricted to Jimma Zone districts and may not be generalizable to Ethiopia or the Oromiya region.

## 1. Background

Human immunodeficiency virus (HIV) is still a serious problem for global public health. In 2020, 37.7 million people worldwide were estimated to be HIV-positive [1]; of these, two-thirds resided in sub-Saharan Africa (SSA). In the same year, about 680,000 people died from acquired immunodeficiency syndrome (AIDS)-related illness worldwide. Of these 45.59% and 22.07% were in eastern and southern Africa, and western and central Africa, respectively [1]. In 2020, Ethiopia saw 670,000 new HIV infections and 12,000 AIDS-related deaths [2].

Although, Ethiopia is on the right track to achieve the United Nation (UN) General Assembly 2016 commitments, i.e., “90–90–90” targets, which seek to ensure that 90% of people living with HIV know their HIV status, 90% of people who know their HIV-positive status in Ethiopia are accessing treatment, and 90% of people on treatment have suppressed viral loads by 2020 [3], were not achieved as expected [4]. Ethiopia remains one of the 30 nations with a high burden of HIV for the years 2015 to 2020, despite the Government’s efforts, including the expansion of integrated tuberculosis and HIV services to the tertiary, secondary, and primary healthcare levels [5]. A report compiled by the Central Statistical Agency (CSA) of Ethiopia using the 2016 Demography and Health Survey data [6] showed that the prevalence of HIV varied by geographic region. The survey also showed that age and socioeconomic status had an impact on HIV prevalence. Studies have also revealed that HIV prevalence varies between SSA’s countries according to factors such as sex, age and place of residence [7,8]. According to our earlier study applying a generalized linear mixed model (GLMM) [9], the likelihood of contracting HIV was also substantially associated with age, gender, marital status, level of education, type of profession, religion, location, and condom use during sexual activity. The results of this study also showed that, in Jimma Zone, there was heterogeneity in the prevalence of HIV infection, with some districts having higher and some districts having lower prevalence rates than the average prevalence rate of Jimma Zone [9]. Furthermore, a district-specific random effect was added to the GLMM to quantify the variation in HIV prevalence caused by differences in districts. Since the random effects are assumed to be independent, it implies that no spatial correlation exists—this is an implausible assumption considering that there is spatial autocorrelation in the country’s HIV prevalence distribution (see e.g., [10,11]).

To develop an efficient intervention approach for reducing the rate of HIV transmission, and to mitigate the HIV epidemic, knowledge of the factors associated with transmission of HIV and identification of the districts in the regions that need priority for interventions are needed [12]. The latter further requires good understanding of the spatial patterns of HIV epidemics at the lower level geographical units of the country [13]. As discussed above, the distribution of HIV in Ethiopia is heterogenous and the reported regional and national estimates of HIV prevalence may mask this heterogeneity within the country. Therefore, when reliable data for lower-level geographic units for HIV diagnosis with potential risk factors are available, assessing the risk factors and spatial components will be vital for any targeted interventions. The aims of the present study, therefore, were to model the district-level HIV prevalence in Jimma Zone, Oromiya region of Ethiopia, with the use of a Bayesian hierarchical spatial modelling approach, as well as to assess the effects of patient characteristics on HIV prevalence. The benefit of Bayesian hierarchical spatial modelling is its ability to incorporate spatial, i.e., neighbourhood, information, on the variable of interest. The available studies related to HIV include only public health facilities in Jimma town (see e.g., [14,15,16,17,18]), while the study by Bedaso and Debusho [9] does not deal with the spatial distribution of HIV prevalence. Therefore, there is no or little information that shows the spatial distribution of HIV prevalence at district-level in Jimma Zone.

## 2. Methods

### 2.1. Study Area and Data

The public health center offices in Jimma Zone districts provided the data for this study. Detailed descriptions of the data including summary statistics are reported in [9]. The data collection was carried out from September 2018 to August 2019 from patients who visited government-owned public health facilities, such as health centres and hospitals, in 22 districts of Jimma Zone. The inclusion criteria for this study comprised patients who had to be at least 15 years old, consented to an HIV test, and had signed a consent form. Data on the patient’s age, HIV status, marital status, educational level, condom use, religion, occupation, and place of residence were collected from the patient’s registry [9]. In this study, the ratio of the number of HIV positive test results in a district to the total number of patients who were tested for HIV in the same district was called “HIV prevalence” for convenience.

### 2.2. Statistical Analyses

#### 2.2.1. Spatial Clustering

Spatial analysis was conducted to identify the geographical clustering of HIV prevalence. For this analysis, the district boundaries were geo-referenced and were connected to the district HIV prevalence data. Then, using ArcGIS software version 10.3 [19], choropleth maps were created for visualization.

The global spatial autocorrelation of HIV was investigated using the global Moran’s *I* statistic [20]. This statistic was applied to assess the presence, direction and strength of spatial autocorrelation over Jimma Zone. The Moran’s I statistic shows positive spatial autocorrelation when values in neighbouring districts are comparable or are spatially grouped. However, when adjacent districts tend to have disparate values or are spatially scattered, it suggests that there is a negative spatial autocorrelation [21,22], and when some neighbouring pairs’ deviations are in the same direction and others are in the opposite direction, it approaches zero [23].

To distinguish between a hotspot and a coldspot in a spatial cluster, the Getis–Ord $G_{i}^{*}$ local statistic can be used instead of the local Moran’s *I* index [24]. In this study, we used the Getis–Ord $G_{i}^{*}$ statistic to determine districts where HIV prevalence was highest (the hotspots) and lowest (the coldspots). Unlike the local Moran’s *I* statistic, the Getis–Ord $G_{i}^{*}$ statistic computed for each district is readily expressed in terms of *z*-scores of the standard normal distribution [25], hence enabling a clearer understanding of statistical significance. Moreover, multiple confidence intervals can be shown for it on a map which are quite appealing to the eye.

#### 2.2.2. Bayesian Hierarchical Spatial Models

Spatial data have the property of spatial dependence [26]. For the present study, this property could be considered as follows: a patient in a district, say $s_{j}$ of Jimma Zone, carries some information about HIV that is observed in districts which are close to $s_{j}$. A spatial correlation analysis of the previous section can be applied to check if this property is evident for any given data. Let an outcome variable $y_{ij}$ denote the *i*th patient HIV test result in the $s_{j}$th district with probability $p_{ij}$, where $y_{ij}=1$ for a patient tested positive and $y_{ij}=0$ for a patient tested negative; $i=1,\ldots ,n_{j}$, j=1,…,22 and $n_{j}$ are the number of patients in the $s_{j}$ district who visited government-owned public health facilities between September 2018 and August 2019. Since the outcome variable had a binary outcome, it was fitted using a Bayesian hierarchical spatial smoothing logistic regression model. The model is given by
$$\eta_{ij}=g\left(p_{ij}\right)=x_{ij}^{\prime }\;fi+u_{j}+v_{j},\;\;i=1,\ldots ,n_{j};\;\;j=1,\ldots ,22;$$
where g(·) is the link function, $x_{ij}=\left(1,x_{1ij},\ldots ,x_{pij}\right)$ is the vector of *p* covariates measured on the *i* patient in the $s_{j}$ district, fi is the vector of regression coefficients including the intercept, $u_{j}$ is the unstructured random component, which is assumed to follow a normal distribution, and $u_{j}∼N\left(0,\sigma_{u}^{2}\right)$ and $v_{j}$ are the structured spatial random components for district $s_{j}$.

Information-sharing between districts, in the Bayesian hierarchical approach, is based on prior assumptions that we place on the spatial dependence structure of the district-specific parameters. In this study, we considered four spatial models (the intrinsic conditional autoregressive (ICAR) [27], the proper CAR (pCAR), Besag, York and Mollié (BYM) [27] and the Leroux conditional autoregressive (LCAR) [28] models) containing spatially structured random effects, which used a Bayesian smoothing conditional autoregressive (CAR) structure [27,29]. A CAR model assumes a form of local spatial continuity, so that neighbouring districts have similar residual risks after accounting for fixed effects [30].

Prior distributions must be specified for each of the model’s random components in a Bayesian method. In this study, non-informative priors were chosen for regression coefficients, or variances of random effects, because there was no empirical data on these model parameters [31]. Specifically, N(0,1000) was specified as a non-informative prior for each regression coefficient. Related to the variances in the four spatial models, the default minimally informative priors were specified in the log of the precision parameters [32].

The integrated nested Laplace approximation (INLA) numerical method [33] was used to fit the Bayesian hierarchical spatial models to the HIV prevalence data. A spatial weight matrix was used to establish the spatial relationships between the districts and the Queen’s contiguity was used to define neighborhoods. Neighboring districts are those that share boundaries or a common vertex. This was defined using the nb2listw function from the spdep [34] R package. The inla() function from the R-INLA package [35] was used to perform the Bayesian analysis. The fitted models were compared using the deviance information criterion (DIC) [36], the effective number of parameters (pD), the mean deviance ($\bar{D}$) and the widely applicable information criterion (WAIC) [37]. The model with the lowest DIC, pD, $\bar{D}$ and WAIC was considered as the best-fit model.

### 2.3. Ethical Consideration

The Postgraduate Research Office of the College of Natural Sciences at Jimma University granted permission for the investigation. Additionally, the Jimma Zone Ministry of Health Office gave their consent for the study data to be used for the investigation. The confidentiality of the patients was guaranteed because the data that the authors were given had no personal identifiers.

## 3. Results

The data set included the 8440 patients who underwent HIV testing between September 2018 and August 2019 in the government clinics in 22 Districts of Jimma Zone. During this period, the overall HIV prevalence among patients who underwent testing at government health service facilities in Jimma Zone was 24.3% for men and 22.1% for women. The raw or unsmoothed prevalence was also varied among districts (ranges between 40–50%), age groups, marital status, education level, type of occupation, religious group and among residential areas. The geographical distribution of the unsmoothed HIV prevalence in Jimma Zone at district level is presented in Figure 1.

The unsmoothed or raw HIV prevalence plot in Figure 1 shows that a high concentration of HIV distribution was observed in Guma, Gomma, Agaro, Limu Kosa and Chora Botor districts.

### 3.1. Spatial Clustering of HIV Prevalence in Jimma Zone

There was evidence of spatial clustering of HIV infection in Jimma Zone with a global Moran’s *I* value 0.457 (*p*-value = 0.0017). The positive global Moran’s *I* suggests that the HIV prevalence in any two spatial neighbouring districts tended to have similar prevalence of HIV. The global Moran’s *I* statistic indicates whether there is spatial aggregation of HIV, but does not, however, allow for the identification of districts’ classification according to the level of their significance. Therefore, to identify high cluster (hotspot) and low cluster (coldspot) districts for HIV prevalence, we applied the Getis–Ord $G_{i}^{*}$ statistic. A map showing the distribution of spatial clusters of HIV prevalence is presented in Figure 2. The map shows that three districts, namely, Agaro, Gomma and Nono Benja, were identified as hotspots with a 95% confidence level, and two districts, namely Mancho and Omo Beyam, as coldspots, with 95% and 90% confidence levels, respectively.

### 3.2. Patient Characteristics Associated with Clustering of HIV Prevalence

To identify patient demographic and other characteristics associated with HIV prevalence, the study data were subjected to a Bayesian hierarchical spatial multivariate logistic regression model. Before including the covariates in the model, the variance inflation factor (VIF) was used to test for multicollinearity. None of the VIFs were above 5, indicating that the collinearity was not significant enough to influence the statistical inference. In the four fitted models, all the patient’s characteristics were statistically significant at a 5% level of significance. Hence, to select an appropriate spatial model for the data, we applied model selection criteria. Table 1 displays the model selection criteria, DIC, pD, $\bar{D}$ and WAIC for the models fitted to the data using different spatial models. The DIC, pD, $\bar{D}$ and WAIC of the ICAR spatial model were marginally smaller than those of the other models, indicating that it should be the model of choice for the data. As a result, we solely present results based on the ICAR model in the sections that follow.

In Table 2, the adjusted odds ratios (aOR) and related 95% credible intervals (CI) for patient characteristics from the fitted Bayesian hierarchical spatial multivariate logistic regression model with spatially structured random effects applying the ICAR model are displayed. All the patient’s characteristics were strongly associated with HIV prevalence. In comparison to male patients, HIV prevalence was 13.8% lower in female patients (aOR = 0.862, 95% CI: 0.751–0.988). Patients aged 15 to 19 and 25 to 49 had an infection prevalence that was 1.776 and 1.016 times higher than that of patients aged 50 or older, respectively (Table 2). The odds of HIV infection were, however, 18.5% (aOR = 0.815, 95% CI: 0.653–1.016) lower for patients in the age group of 20 to 24 than patients who were 50 years of age or older. Patients with primary, secondary, and higher education levels were 1.546 (aOR = 1.546, 95% CI: 1.307–1.828), 1.210 (aOR = 1.210, 95% CI: 0.996–1.470), and 1.233 (aOR = 1.233, 95% CI: 0.922–1.651) times more likely to have HIV infection than patients with no formal education, respectively. The increased risk of HIV infection was statistically insignificant for secondary and superior education levels, though the 95% CI credible intervals included one. Married, divorced and widowed patients were 1.592 (aOR = 1.592, 95% CI: 1.302–1.948), 1.483 (aOR = 1.483, 95% CI: 1.188–1.852) and 1.514 (aOR = 1.514, 95% CI: 1.176–1.950) times more likely to have HIV infection, respectively, compared to unmarried or single patients.

Patients who worked as daily laborers, farmers, government employees, or merchants had HIV infection rates that were 1.212, 1.252, 1.221, and 1.188 times higher, respectively, than patients who had no occupation (aOR = 1.212, 95% CI: 0.992–1.478 for daily laborers; 1.252, 1.004–1.560 for farmers; 1.221, 95% CI: 0.874–1.708 for government employees; and aOR = 1.188, 95% CI: 0.939–1.504 for merchants). Only patients who were farmers, however, showed a statistically significant rise in HIV infection risks. Orthodox and Protestant patients had odds of HIV infection that were 12.1 and 29.8 per cent lower than Muslim patients (aOR = 0.879, 95% CI: 0.727–1.064 for orthodox; aOR = 0.702, 95% CI: 0.603–0.818 for Protestant). Patients who lived in urban areas had 1.213 times higher risk of HIV infection than those who lived in rural areas (aOR = 1.213, 95% CI: 1.057–1.392). Patients who had sexual intercourse without a condom were 71.008 times more likely to be infected compared to those who used a condom during sexual intercourse (aOR = 71.008, 95% CI: 61.284–82.513).

### 3.3. Spatial Analysis

The smoothed HIV prevalence rates of districts are displayed in Figure 3. In this figure the prevalence intervals of Figure 1 were used to provide a comparison with the unsmoothed prevalence rates. For most districts, the general spatial patterns seen in the unsmoothed map of HIV prevalence in Figure 1 were changed compared to the spatial patterns of smoothed HIV prevalence. However, the concentration of high HIV prevalence districts for the unsmoothed estimates were still part of districts with high HIV prevalence for the smoothed estimates.

The spatially structured residuals of the fitted model were examined for spatial autocorrelation. After considering the patient characteristics, the global Moran’s *I* = 0.239 (*p*-value = 0.0510) for spatially structured residuals was marginally non-significant at the 5% significance level. The map for spatial clustering of these residuals is given in Figure 4. The plot shows that there were two hotspots, Jimma special town and Mana district, which were significant at 95% and one lower cluster or coldspot (Setema district). The two hotspots showed that there was spatial clustering of residuals because the global Moran’s *I* was positive. These results indicate that not all the district level heterogeneities in HIV prevalence in Jimma Zone were explained by patient characteristics for the study data.

## 4. Discussion

The present study identified that there was variation in HIV prevalence at district level in the study period in Jimma Zone. The spatial distribution of the raw or unsmoothed HIV prevalence by districts showed that high HIV infection among tested patients was observed in Guma, Gomma, Agaro, Limu Kosa and Chora Botor districts, and lower HIV infection in Dedo, Omo Nada, Sekoru, Shebe Senbo and Tiro Afeta districts. The spatial clustering analysis also identified that Agaro, Gomma and Nono Benja districts were hotspots for HIV infection.

The global Moran’s *I* statistic was positive, indicating that the prevalence of HIV trends were similar in neighboring districts. The results from LISA using the Getis–Ord $G_{i}^{*}$ statistic showed that the HIV prevalence was spatially clustered, i.e., there were districts, Agaro, Gomma and Nono Benja, identified as hotspots. The hotspot of Agaro may be because the district is one of the most important coffee trading centres in the western part of Ethiopia and the main road from Bedele town (the town being the headquarters for the Bedele Brewery) to Jimma town crosses Agaro town; therefore, the town serves as a transportation route for long-distance or truck drivers. In general, these localities have a larger concentration of commercial sex workers [38,39,40], and towns and urban areas in general have a higher risk of HIV infection [41,42] due to increased population movement brought on by labour, migration, and trade. Gomma district has various industries that attract people for job seeking, such as coffee hulling, coffee pulping mills and sawmills. As a result, the hotspot in this district may be associated with people moving between or within districts in search of employment or better living conditions. A study undertaken in sub-Saharan Africa has identified the high mobility of the population in search of better living conditions as a potential catalyst for the HIV epidemic [43]. The Nono Benja district boarders with the southwest Shewa Zone of Oromiya region which has well developed cities, such as Waliso town; this may attract people from Nono Benja for job searching. If those who migrated to this town are infected by HIV, when they return to Nono they could transfer the virus to their partners in unsafe sexual intercourse. About 55% of patients in this district had sexual intercourse without a condom before the HIV test. The spatial heterogeneity in HIV infection at district level in Jimma Zone suggests that targeted interventions by authorities may be needed to prevent HIV transmission, especially in hotspot districts.

Using the DIC, pD, $\bar{D}$ and WAIC, we have shown that the ICAR spatial model had a slightly better fit. For most districts, the general spatial patterns seen in the raw or unsmoothed map of HIV prevalence were changed compared to the spatial patterns of smoothed HIV prevalence. However, the concentration of high HIV prevalence districts for raw estimates was still part of those districts with high HIV prevalence for the smoothed estimates. Generally, the unsmoothed prevalence is misleading when the geographical areas are small. Although the smoothed HIV prevalence map shows improvement over the unsmoothed map, the interpretation of the map needs caution because there could be confounding factors, differences in population sizes among districts, and district factors. In the Bayesian hierarchical spatial model, we used an adjacency weight matrix to determine the number of neighbours each district had, but some districts in Jimma Zone border with districts in other zones in the Oromiya region and in other regions, e.g., the Southern Nations, Nationalities, and Peoples’ regions. However, there were no data available for these zones; therefore, those districts in Jimma Zone bordering other districts from other zones had missing neighbours. Thus, this missingness could introduce some bias when averaging HIV prevalence or estimating spatial effects because the variances of special effects are functions of the number of neighbours. However more reliable estimates of prevalence for small areas can be obtained by the Bayesian hierarchical spatial model, which is used in this paper, as it allows borrowing information from other neighbouring districts [44].

The results from fitting the Bayesian hierarchical spatial multivariate logistic regression considering the spatially structured random effect revealed that gender, age, marital status, education level, type of profession, religion, place of residence, and condom use during sex all significantly influenced the likelihood of contracting HIV. These findings concur with earlier research [45,46,47]. Further, the results from spatial clustering analyses on the spatially structured residuals suggested that, for the study data, not all the heterogeneity in HIV prevalence in Jimma Zone was explained by patient characteristics. The existing studies on HIV infection in Jimma Zone were based on data collected from public health facilities in Jimma town (see, e.g., [14,15,16,17,18]) and they did not consider the spatial distribution of HIV prevalence. The identification of cluster (hotspot) districts may indicate an expanded epidemiological view of the districts with high risk of HIV transmission in Jimma Zone. The results further revealed districts that require targeted interventions and consolidation of various surveillance measures.

Despite these strengths, the study’s shortcomings are acknowledged by the authors. According to other studies, the prevalence of HIV infection is related to a person’s HIV knowledge, clinical characteristics, such as STIs [46,47], and the characteristics of his/her sexual behavior, such as his/her age at first sex, condom use at first sex, number of sex partners, and frequency of HIV testing. However, this information was not available in the clinic patient’s register where the study data were collected; therefore, these variables were not introduced in the analysis. In addition, in this paper, we used the HIV test result data as a surrogate measure of the general population of HIV-infected persons in Jimma Zone. Since these were data collected from specific sub-populations who sought treatment and care from nearby health facilities, they represent populations around the facilities. Hence, care should be taken when interpreting the current results. The results are also limited to Jimma Zone districts and may not necessarily generalize to the Oromiya region or Ethiopia.

## 5. Conclusions

In this study, we investigated spatial clustering of the prevalence of HIV in Jimma Zone districts, Oromiya region of Ethiopia, and, at the individual level, the effects of patient characteristics on HIV prevalence, using a Bayesian hierarchical spatial modelling approach. Bayesian hierarchical spatial models were applied using various spatial models and the best model was selected using the DIC, pD, $\bar{D}$ and WAIC.

The identification of hotspot districts and the spatial dynamics of HIV infection in Jimma Zone at district level may allow health policymakers in the zone or Oromiya region or at national level to plan localized interventions tailored to at-risk individuals to control the transmission of HIV. The methods applied in this paper can be extended to investigate the zones HIV prevalence relative to the prevalence at region level or a region’s HIV prevalence relative to the prevalence at national level.

## Figures and Tables

**Figure 1 diseases-11-00046-f001:**
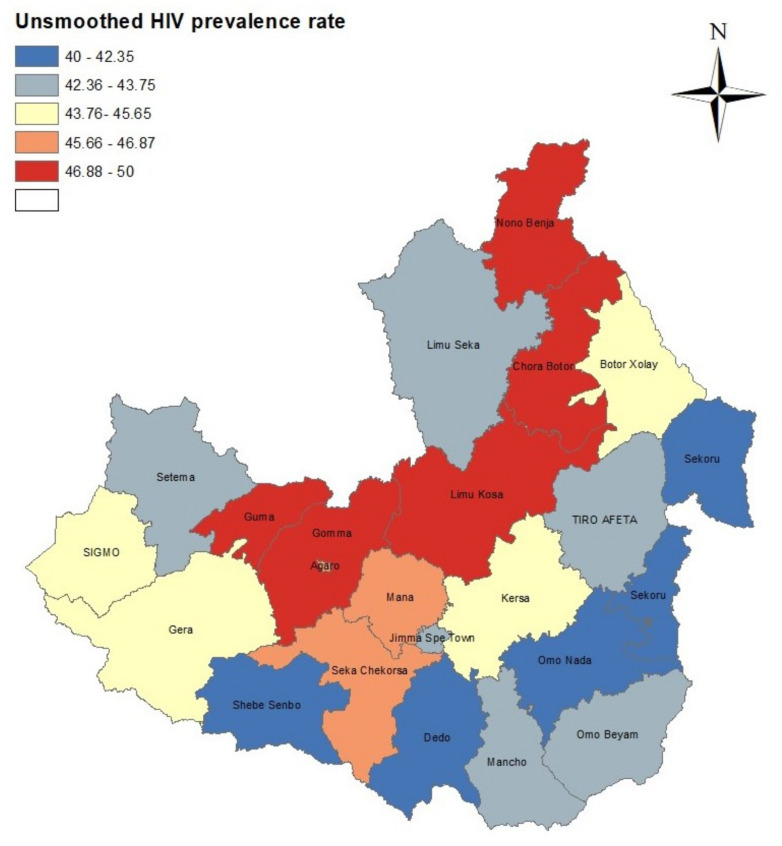
Geographical distribution of raw (unsmoothed) HIV prevalence in Jimma Zone at district level.

**Figure 2 diseases-11-00046-f002:**
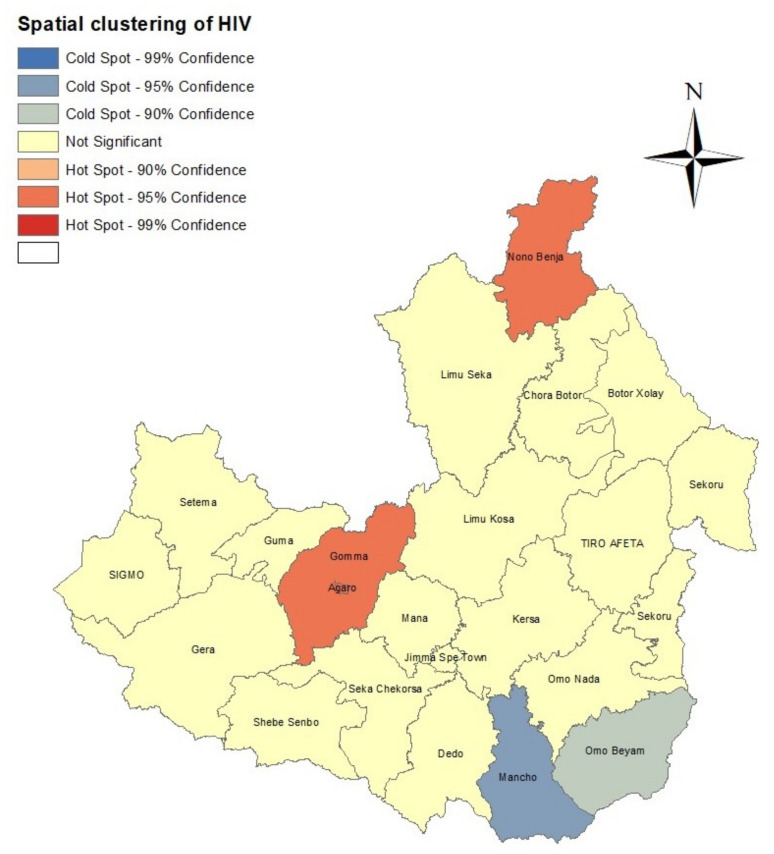
Spatial clustering of HIV prevalence in Jimma Zone at district level based on the Getis–Ord $G_{i}^{*}$ statistic.

**Figure 3 diseases-11-00046-f003:**
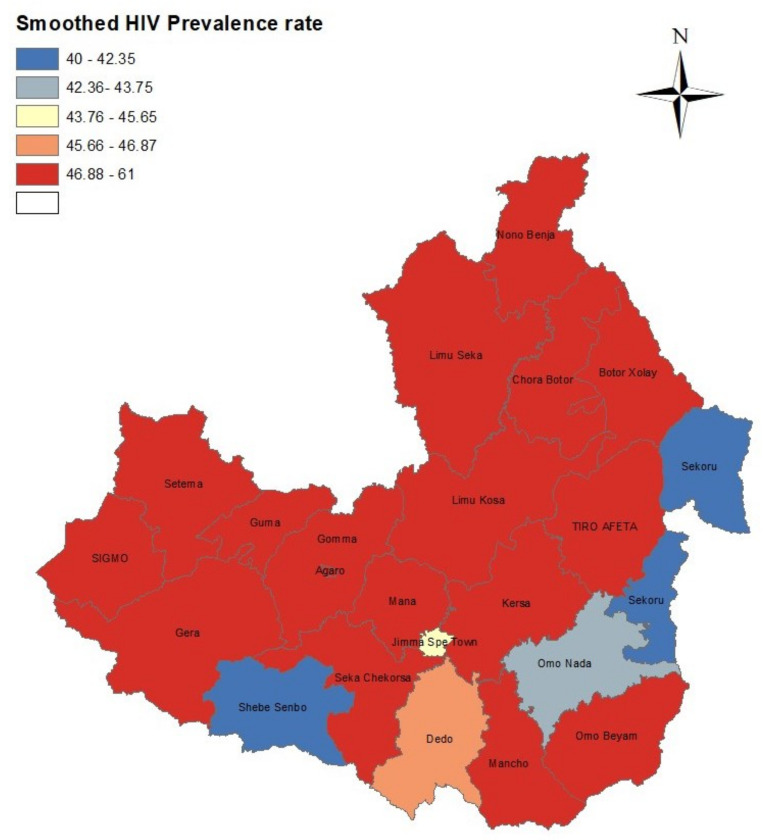
Geographical distribution of posterior means or smoothed HIV prevalence.

**Figure 4 diseases-11-00046-f004:**
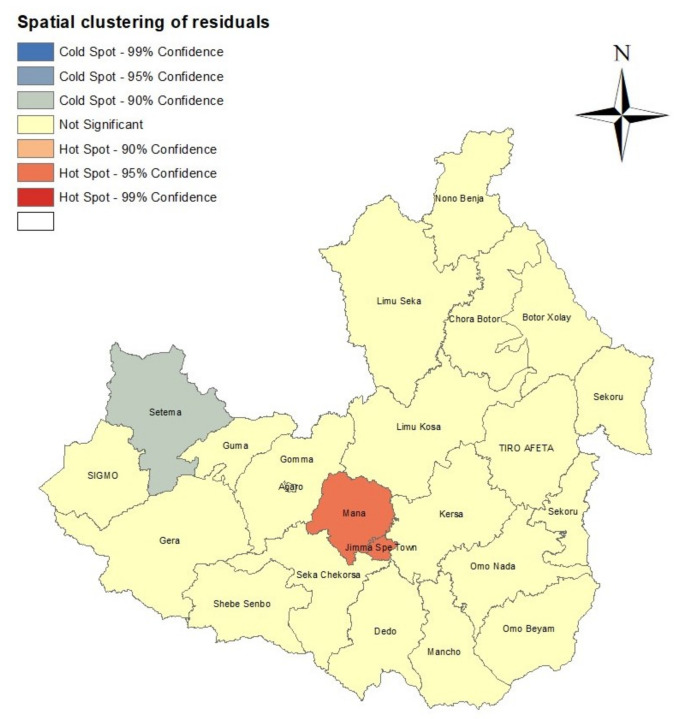
Spatial clustering of spatially structured residuals using the Getis–Ord statistic.

**Table 1 diseases-11-00046-t001:** Summary of DIC, pD, $\bar{D}$ and WAIC values for different spatial models used to fit the Bayesian hierarchical spatial models to HIV prevalence data.

Spatial Model	DIC	pD	$\bar{D}$	WAIC
ICAR	6095.732	33.39138	6062.341	6095.987
pCAR	6098.995	34.05564	6064.939	6099.551
BYM	6096.888	33.48582	6063.403	6097.273
LCAR	6096.986	33.77660	6063.210	6097.409

**Table 2 diseases-11-00046-t002:** Adjusted odds ratios (95% credible interval for parameters in the Bayesian spatial model fitted to the HIV data using ICAR model.

Coefficients	Adjusted Odds Ratio
	(95% Credible Interval)
Intercept	0.039 (0.028, 0.055)
Age (ref: ≥50)	
15–19	1.776 (1.371, 2.302)
20–24	0.815 (0.653, 1.016)
25–49	1.016 (0.822, 1.256)
Gender (ref: Male)	
Female	0.862 (0.751, 0.988)
Marital status (ref: Single)	
Married	1.592 (1.302, 1.948)
Divorced	1.483 (1.188, 1.852)
Widowed	1.514 (1.176, 1.950)
Education level (ref: No education)	
Primary	1.546 (1.307, 1.828)
Secondary	1.210 (0.996, 1.470)
Superior	1.233 (0.922, 1.651)
Occupation (ref: No job)	
Daily laborer	1.212 (0.992, 1.478)
Farmer	1.252 (1.004, 1.560)
Government employee	1.221 (0.874, 1.708)
Merchant	1.188 (0.939, 1.504)
Religion (ref: Muslim)	
Orthodox	0.879 (0.727, 1.064)
Protestant	0.702 (0.603, 0.818)
Residence (ref: Rural)	
Urban	1.213 (1.057, 1.392)
Condom use (ref: Yes)	
No	71.008 (61.284, 82.513)
	Median
*Random effects*	(95% Credible interval)
Structured variance	1.647 (1.347, 2.153)

## Data Availability

The data that support the findings of this study are available from the Jimma Zone Ministry of Health Office but restrictions apply to the availability of these data, which were used under license for the current study and so are not publicly available. Data are however available from the authors upon reasonable request and with permission of the Jimma Zone Ministry of Health Office.

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
