# Peer review of "Bayesian Spatial Modelling of HIV Prevalence in Jimma Zone, Ethiopia"

_diseases, 2023, doi:10.3390/diseases11010046_

Round 1

Reviewer 1 Report

I think it is overall nice written but there is a need to improve English language and style 

Author Response

Comments and Suggestions for Authors

I think it is overall nice written but there is a need to improve English language and style.

Authors’ response:

We thank the reviewer for this comment. We have carefully read the manuscript and made changes in the language wherever it was necessary, and these were marked in RED in the revised version of the manuscript.

Reviewer 2 Report

The authors have presented very good work and the methods proposed are robust and sheds light on HIV burden and how targeted interventions can implement. The authors should, however, consider the following;

1.  On page 4, the authors talk of crude HIV prevalence, it is not clear how they arrived at the crude prevalence rates. My understanding is that data related to those who presented themselves to government facilities and tested. Prevalence would imply that there was a population level assessment. My understanding is of those tested how many or what % turned positive and cannot be synonymous with prevalence.

2. Whereas the results are very insightful, the authors need to provide or control for population size in each of the districts i.e. Getting the crude rates for each district. 

3. Clarity on some of the spatial characteristics which could explain the difference i.e. whether a district is rural or urban, the economic status in the district etc. How did they integrate this in the analysis? it would be key to see how they have factored this in their analysis

4. In the introductory section the authors should explain the burden of HIV in Jimma Zone and how does this differ from other zones as well. The cases in the Zone account for what % of the cases in the Country.

5.  Would the methods of this study be used at national scale? If yes, this should be one of the recommendations of your study.

Author Response

Reviewer 2 Comments

Comments and Suggestions for Authors

The authors have presented very good work and the methods proposed are robust and sheds light on HIV burden and how targeted interventions can implement. The authors should, however, consider the following;

  1. On page 4, the authors talk of crude HIV prevalence, it is not clear how they arrived at the crude prevalence rates. My understanding is that data related to those who presented themselves to government facilities and tested. Prevalence would imply that there was a population level assessment. My understanding is of those tested how many or what % turned positive and cannot be synonymous with prevalence.

Authors’ response:

We thank the reviewer for this comment. To avoid confusion, we add the statement “In this study, the ratio of number of HIV positive test results in a district to the total number of patients who were tested for HIV in the same district was called "HIV prevalence" for convenience.” In Lines 84-86 (page 2). Furthermore, the word “crude” replaced by raw or unsmoothed throughout the manuscript.

  1. Whereas the results are very insightful, the authors need to provide or control for population size in each of the districts i.e. Getting the crude rates for each district.

Authors’ response:

We agree with the reviewer. However, for us to provide the crude rates, we need districts population sizes by gender and age, at least, for both rural and urban areas. Honestly, we have looked for this information at the beginning of our research but could not find the data. As we have mentioned above, we have removed the word “crude” from the manuscript as this may also raise similar question by readers.

  1. Clarity on some of the spatial characteristics which could explain the difference i.e. whether a district is rural or urban, the economic status in the district etc. How did they integrate this in the analysis? it would be key to see how they have factored this in their analysis

Authors’ response:

As our study based on individual patients’ data, we have included available information related to the issues raised by the review, specifically occupation (which is directly related to patient’s economic status), and place of residence (urban or rural) in the fitted model. In the manuscript, we have examined the spatially structured residuals of the fitted model for spatial autocorrelation after considering the patient characteristics, including the above two variables. The authors believe that the issue raised by the Reviewer are already included in the analysis.

  1. In the introductory section the authors should explain the burden of HIV in Jimma Zone and how does this differ from other zones as well. The cases in the Zone account for what % of the cases in the Country.

Authors’ response:

Most of the available studies include only public health facilities in Jimma town, for example, Girma, Assegid and Gezahegn (2021); Yideg et al. (2019); Weldsilase et al. (2018); Kassahun et al. (2018) and Tesfaye (2018). However, information that shows the comparison of HIV distribution over district-to-district levels in Jimma Zone or zone-to zone levels are not available. Even the Ethiopia Demographic and Health Survey, i.e., EDHS 2016 report (CSA & ICH, 2017) does not contain district or zone prevalence estimates. Therefore, we could not address the above comment in the manuscript.

References

  • Central Statistical Agency (CSA) [Ethiopia] and ICF (2017). Ethiopia Demographic and Health Survey 2016. Addis Ababa, Ethiopia. http://dhsprogram.com/pubs/pdf/FR328/FR328.pd
  • Tesfaye, T., Darega, J., Belachew, T. et al. (2018). HIV positive sero-status disclosure and its determinants among people living with HIV /AIDS following ART clinic in Jimma University Specialized Hospital, Southwest Ethiopia: a facility- based cross-sectional study. Arch Public Health, 76 (1). https://doi.org/10.1186/s13690-017-0251-3
  • Kassahun G, Tenaw Z, Belachew T, Sinaga M (2018). Determinants and Status of HIV Disclosure among Reproductive Age Women on Antiretroviral Therapy at Three Health Facilities in Jimma Town, Ethiopia, 2017. Health Sci J. Vol. 12 No. 2: 558.
  • Yetnayet Abebe Weldsilase, Melaku Haile Likka, Tolossa Wakayo, Mulusew Gerbaba (2018). Health-Related Quality of Life and Associated Factors among Women on Antiretroviral Therapy in Health Facilities of Jimma Town, Southwest Ethiopia. Advances in Public Health, vol. 2018, Article ID 5965343. https://doi.org/10.1155/2018/5965343
  • Yideg Yitbarek, G., Mossie Ayana, A., Bariso Gare, M. and Garedew Woldeamanuel, G. (2019). Prevalence of Cognitive Impairment and Its Predictors among HIV/AIDS Patients on Antiretroviral Therapy in Jimma University Medical Center, Southwest Ethiopia. Psychiatry J. https://doi.org/10.1155/2019/8306823
  • Girma, D., Assegid, S. and Gezahegn, Y. (2021). Depression and associated factors among HIV-positive youths attending antiretroviral therapy clinics in Jimma town, southwest Ethiopia. PLOSS ONE, 16 (1); https://doi.org/10.1371/journal.pone.0244879

  1. Would the methods of this study be used at national scale? If yes, this should be one of the recommendations of your study.

Authors’ response:

Thank you for the comments. Yes, the methods used in the manuscript applicable for either regional or national level. We add the following “The methods applied in this paper can be extended to investigate the zones HIV prevalence relative to the prevalence at a region level or the regions HIV prevalence relative to the prevalence at national level.” At the end of the conclusion section as a recommendation.

Reviewer 3 Report

This is a well-written, well-organized and well-illustrated paper. It presents the results of original research and makes a valuable contribution to knowledge.

Author Response

Comments and Suggestions for Authors

This is a well-written, well-organized and well-illustrated paper. It presents the results of original research and makes a valuable contribution to knowledge.

Authors’ response:

We are very happy to receive the above motivating words from the Reviewer.

We thoroughly reviewed the manuscript to check for possible language and spelling errors, made corrections to those we identified. These were indicated in red colour in the revised version of the manuscript. 

Reviewer 4 Report

This is a good paper, but I have some questions and suggestions.

All my comments are relative to the statistical analysis, because the other sections are relatively short and are not my field of expertise either. 

Regarding the model, it could help to see it, even if the journal is about medicine, to dedicate 3 lines to write the model could help to see which "variables" are studied... because if in another paragraph it talks about Collinearity, that means that the model has different "X "s.

On the other hand, I find it strange to see a Disease Mapping model, and then a single OR table, that is to say... I show the map but then I give you a single global information... so what is the map for? Are they two different analyses? I don't quite understand this.

The DIC has been used for the models, when possibly the WAIC gives better results and differentiates more between models. But, what I do not understand is to choose the model with DIC by changing the priors... that does not make sense... the priors may or may not affect the model, but they are not part of the model selection, the model selection is done by removing or putting variables but changing the priors.

Please thin about all these things and rewire the paper.

Author Response

Comments and Suggestions for Authors

This is a good paper, but I have some questions and suggestions.

All my comments are relative to the statistical analysis, because the other sections are relatively short and are not my field of expertise either.

  • Regarding the model, it could help to see it, even if the journal is about medicine, to dedicate 3 lines to write the model could help to see which "variables" are studied... because if in another paragraph it talks about Collinearity, that means that the model has different "X "s.

Authors’ response:

Thank you for the suggestion. We have added the model in the “Bayesian hierarchical spatial models” section, please see Lines 109-113 on page 3.

  • On the other hand, I find it strange to see a Disease Mapping model, and then a single OR table, that is to say... I show the map but then I give you a single global information... so what is the map for? Are they two different analyses? I don't quite understand this.

Authors’ response:

Please note that we have fitted four Bayesian hierarchical spatial smoothing logistic regression models and selected one using model selection criteria. Further, we have mentioned in “Patient characteristics associated with clustering of HIV prevalence” subsection that we are going to report the results from the selected model only. The interpretation / discussion of a logistic regression usually done by using odds ratio. That is why we have provided a single model. Note also that Figures 3 and 4 were generated from results of the fitted Bayesian hierarchical spatial smoothing logistic regression model (selected model). The smoothed HIV prevalence rates of districts and the spatially structured residuals, which were plotted in Figure 3 and Figure 4, respectively were generated from the fitted model. The addition of the model structure may make the presented analysis results clearer in the revised version of the manuscript.

  • The DIC has been used for the models, when possibly the WAIC gives better results and differentiates more between models. But, what I do not understand is to choose the model with DIC by changing the priors... that does not make sense... the priors may or may not affect the model, but they are not part of the model selection, the model selection is done by removing or putting variables but changing the priors.

Authors’ response:

Thank you for the above comments. We have included the WAIC in the revised version. In the four fitted models, all the covariates were statistically significant. We have used the model selection criteria to select the spatial structure or spatial model which described by four different models. In the revised version of the manuscript, we have clearly indicated why we used the model selection criteria. It was used to select the model that described the spatial structure best. For example, the following sentences were added in Lines 173-176 on pages 4-5.

“In the four fitted models, all the patient's characteristics were statistically significant at 5\% level of significance. Hence, to select appropriate spatial model for the data, we applied the model selection criteria.”

  • Please thin about all these things and rewire the paper.

Authors’ response:

Thank you for the comment. We have made changes in the revised version of the manuscript wherever it was necessary based on the four reviewers, including your comments and the Assistant Editor comments / suggestion.

Round 2

Reviewer 2 Report

The authors have made the required changes. This paper adds to the body on knowledge on understanding HIV along the space and time continuum.

Reviewer 4 Report

Thanks for answering my questions and take into account my suggestions. This version improve previous one.